# Reduction of antisense transcription affects bovine leukemia virus replication and oncogenesis

**Thomas Joris**[1‡], **Thomas Jouant**[1‡], **Jean-Rock Jacques**[1], **Lorian Gouverneur**[1], **Xavier Saintmard**[1], **Lea Vilanova Mañá**[1], **Majeed Jamakhani**[1], **Michal Reichert**[2], **Luc Willems**[1]*

1 Molecular and Cellular Epigenetics, Interdisciplinary Cluster for Applied Genoproteomics (GIGA), Sart-Tilman, Liège, Belgium; Molecular Biology, Teaching and Research Centre (TERRA), Gembloux, Belgium, 2 Department of Pathological Anatomy, National Veterinary Research Institute, Puławy, Poland

‡ These authors share first authorship on this work.
* luc.willems@uliege.be

**Data Availability Statement:** RNA sequencing Data has been uploaded on GEO. The accession number is GSE255715.

## Abstract

In sheep infected with bovine leukemia virus (BLV), transcription of structural, enzymatic, and accessory genes is silenced. However, the BLV provirus transcribes a series of non-coding RNAs that remain undetected by the host immune response. Specifically, three RNAs (AS1-L, AS1-S, and AS2) are consistently expressed from the antisense strand, originating from transcriptional initiation at the 3'-Long Terminal Repeat (LTR). To investigate the role of these non-coding RNAs in viral replication and pathogenesis, a reverse genetics approach was devised, capitalizing on a mechanistic disparity in transcription initiation between the 5' and 3' promoters. A two-nucleotide mutation (GG>TA) in the TFIIB-recognition element (BRE) impaired antisense transcription originating from the 3'-LTR. In the context of the provirus, this 2bp mutation significantly diminished the expression of antisense RNAs, while not notably affecting sense transcription. When inoculated to sheep, the mutated provirus was infectious but exhibited reduced replication levels, shedding light on the role of antisense transcription in vivo. In comparison to lymphoid organs in sheep infected with a wild-type (WT) provirus, the mutant demonstrated alterations in both the spatial distribution and rates of cell proliferation in the lymph nodes and the spleen. Analysis through RNA sequencing and RT-qPCR unveiled an upregulation of the *Hmcn1/*hemicentin-1 gene in B-lymphocytes from sheep infected with the mutated provirus. Further examination via confocal microscopy and immunohistochemistry revealed an increase in the amount of hemicentin-1 protein encoded by *Hmcn1* in peripheral blood mononuclear cells (PBMCs) and lymphoid organs of sheep infected with the mutant. RNA interference targeting *Hmcn1* expression impacted the migration of ovine kidney (OVK) cells *in vitro*. In contrast to the WT, the mutated provirus showed reduced oncogenicity when inoculated into sheep. Collectively, this study underscores the essential role of antisense transcription in BLV replication and pathogenicity. These findings may offer valuable insights into understanding the relevance of antisense transcription in the context of human T-cell leukemia virus (HTLV-1).

**Funding:** This work was supported by the Fonds de la Recherche Scientifique-FNRS grant numbers FNRS PDR T.0261.20, PER H.P055.20, CDR J.0195.21, and J.0194.23 to LW, Télévie grants 7460621 and 7460923 to LW, Université de Liège grant FSR-S-SS-2023/14 to LW, and Fonds Léon Fredericq to LW. The funders played no role in the design of the study, the collection and analysis of data, the decision to publish or the preparation of the manuscript.

**Competing interests:** The authors have declared that no competing interests exist.

## Author summary

In sheep infected with bovine leukemia virus (BLV), the transcription of genes encoding viral proteins is silenced. However, the virus constitutively produces long non-coding RNAs from the opposite strand of the integrated genome. To comprehend their role, we used reverse genetics to alter the BLV sequence, reducing the production of these antisense RNAs. When inoculated to sheep, this modified virus was infectious but could not replicate as well. Compared to the original strain, the number of mutated viruses was reduced in the peripheral blood and the lymphoid organs. Gene expression in B-cells from sheep infected with the mutated virus was altered, particularly affecting the *Hmcn1* gene, which is involved in cell migration. Importantly, the modified virus did not cause leukemia in sheep, indicating the involvement of antisense RNAs in effective cancer development. In summary, our study underscores the significance of these antisense RNAs in the ability of the virus to replicate and induce illness. This understanding may offer insights into similar mechanisms in viruses affecting humans such as HTLV-1.

## Introduction

A series of species including cattle (*Bos taurus*), water buffalo (*Bubalus bubalis*), yak (*Bos grunniens*) and zebu (*Bos taurus indicus*) can be naturally infected with the bovine leukemia virus (BLV) [1–4]. In the bovine species, persistent infection by BLV leads to B-cell lymphoproliferative diseases: persistent lymphocytosis (PL) and leukemia/lymphoma (enzootic bovine leukosis, EBL) [2]. Persistent infection of bovines by BLV is associated with impaired immunity, weight loss and an overall decrease of productivity, which is detrimental to the dairy and meat industries [5,6]. While the implementation of eradication measures successfully resulted in a BLV-free status in European countries [7], EBL prevalence is rising to severe levels elsewhere [8–10]. Although culling of cattle having high proviral loads is possible, vaccination is likely the most effective strategy in regions characterized by a high prevalence [9,11]. Besides prevention, there is no therapeutic option that prevents disease onset, although administration of lysine deacetylase inhibitors effectively reduce proviral loads *in vivo* [12]. Understanding the mechanisms of BLV replication and pathogenesis is thus required to develop novel economically sustainable strategies. In a perspective of the one health concept, BLV infection has been correlated with human cancer, although convincing experimental evidence demonstrating a causal link is still lacking [13,14].

Besides the development of therapeutic and preventive strategies, the BLV model may be useful to better understand the human T-cell leukemia virus (HTLV-1). In a comparative virology approach, both related retroviruses indeed share several characteristics but infect different species. Unfortunately, the long latency period (7–10 years) and the low frequency of disease onset (5–10%) are major limitations of studies based on bovines naturally infected with BLV. However, BLV can also be experimentally inoculated to sheep, inducing almost invariably leukemia/lymphoma after a latency period of 2–3 years [15].

In this study, focus was given on a particular mechanism shared by BLV and HTLV-1: the ability of their long terminal repeats (LTRs) to initiate transcription in the sense and antisense orientations. Transcription directed by the 5'-LTR promoter generates the genomic viral RNA that also encodes the structural and enzymatic genes: *gag* (MA, NC and CA), *pro* (protease) and *pol* (reverse transcriptase and integrase). Alternative splicing further creates the transcript of *env* (SU and TM) as well as other RNAs including *tax*, *rex*, *R3* and *G4*. Transcription from

the antisense strand initiates from a promoter located in the 3'-LTR generating a series of viral RNAs (*as1-L / as1-S / as2* in BLV and *shbz / ushbz* in HTLV-1) [16,17]. The role of the BLV antisense RNAs is still unknown but is predicted to affect epigenetic mechanisms due to the lack of a significant open reading frame and to their major nuclear localization [18]. In contrast, the antisense RNA of HTLV-1 encodes the HBZ protein (HTLV-1 bZIP factor) that has been much better characterized [19–21]. Notably, the HBZ polypeptide counteracts some functions exerted by Tax protein including activation of 5'-LTR-directed transcription and NFκB signaling [19,21,22]. In absence of translated protein, the *hbz* ribonucleic acid prevents the binding of the TATA-binding protein (TBP) to the 5'-LTR chromatin and represses transcription of sense RNAs [23].

Upon infection of a cell by a BLV virion, the process of reverse transcription of the genomic RNA produces identical copies of the 5'-LTR and 3'-LTR. After integration, the provirus can remain silent or be transcribed in both orientations from different promoters located in these terminal sequences. Sense transcription initiates in the U3 region of the 5'-LTR with the binding of TBP to the TATA box, which in turn recruits the RNA polymerase (RNAP) type II (RNAPII) (**Fig 1A**). Upstream of this transcriptional start site, three 21-base pairs (bp) Tax-responsive elements (TxRE) contain imperfectly conserved cyclic-AMP recognition element (CRE) that interact with the CRE-binding protein (CREB) and activation transcription factors 1/2 (ATF1/2) [24,25]. This complex further recruits the CREB-binding protein (CBP), that acetylates the surrounding chromatin. Although other cellular factors interact with the 5'-LTR, the major activator of sense transcription is the viral Tax protein [17]. The process of antisense transcription is initiated in the 3'-LTR at a TATA-less RNAPII core promoter containing a TFIIB-recognition element (BRE) and motif ten/downstream promoter elements (MTE/DPE) (**Fig 1A**). In this study, the mechanistic difference in 5'- and 3'-LTR promoter initiation was exploited to investigate the role of antisense transcription using a reverse genetics approach.

## Results

### A mutation of the BRE decreases antisense transcriptional activity of the LTR promoter

To understand the biological function of BLV antisense transcription, a reverse genetics approach was used to selectively target the antisense promoter located in the 3'-LTR (**Fig 1A**). Therefore, a series of mutations were introduced by site-directed mutagenesis in the BRE and MTE/DPE core elements located in the R-U5 region of the LTR as indicated on **Fig 1B**. The mutations were targeted to the 2 transcription initiation sites of antisense RNAs (A, A1 and A2 in the BRE and B in the MTE/DPE). The GG>TA and GC>TA mutations were designed to disrupt the interaction of the promoter element to the Transcription factor II B (TFIIB). The mutated LTR sequences were inserted into a reporter plasmid containing the Renilla and Firefly luciferase genes cloned in opposite orientations (**Fig 1C**). The different mutants were introduced in bidirectional reporter vectors containing the full-length LTR or a minimal promoter devoid of U3 (R-U5). Upon transfection of human embryonic kidney cells (HEK293FT) with the R-U5 reporter, no Firefly-luciferase activity could be detected above background levels indicating that sense transcription required U3, as expected (**Fig 1D**). Transfection of the R-U5 reporter produced Renilla-luciferase activity that was not significantly affected by the A2/B1 mutations (**Fig 1E**). In contrast, the GG to TA mutation present in the A/A1 reporters reduced Renilla-luciferase activity. A similar effect of the mutations was observed in the presence of the Tax transcriptional activator (pSGTax [26]). In the context of a full-length LTR, the A1 mutation slightly increased expression of Firefly-luciferase in basal conditions (pSG5) and after co-transfection of pSGTax (**Fig 1F**). Compared to the WT, Renilla-luciferase activity was

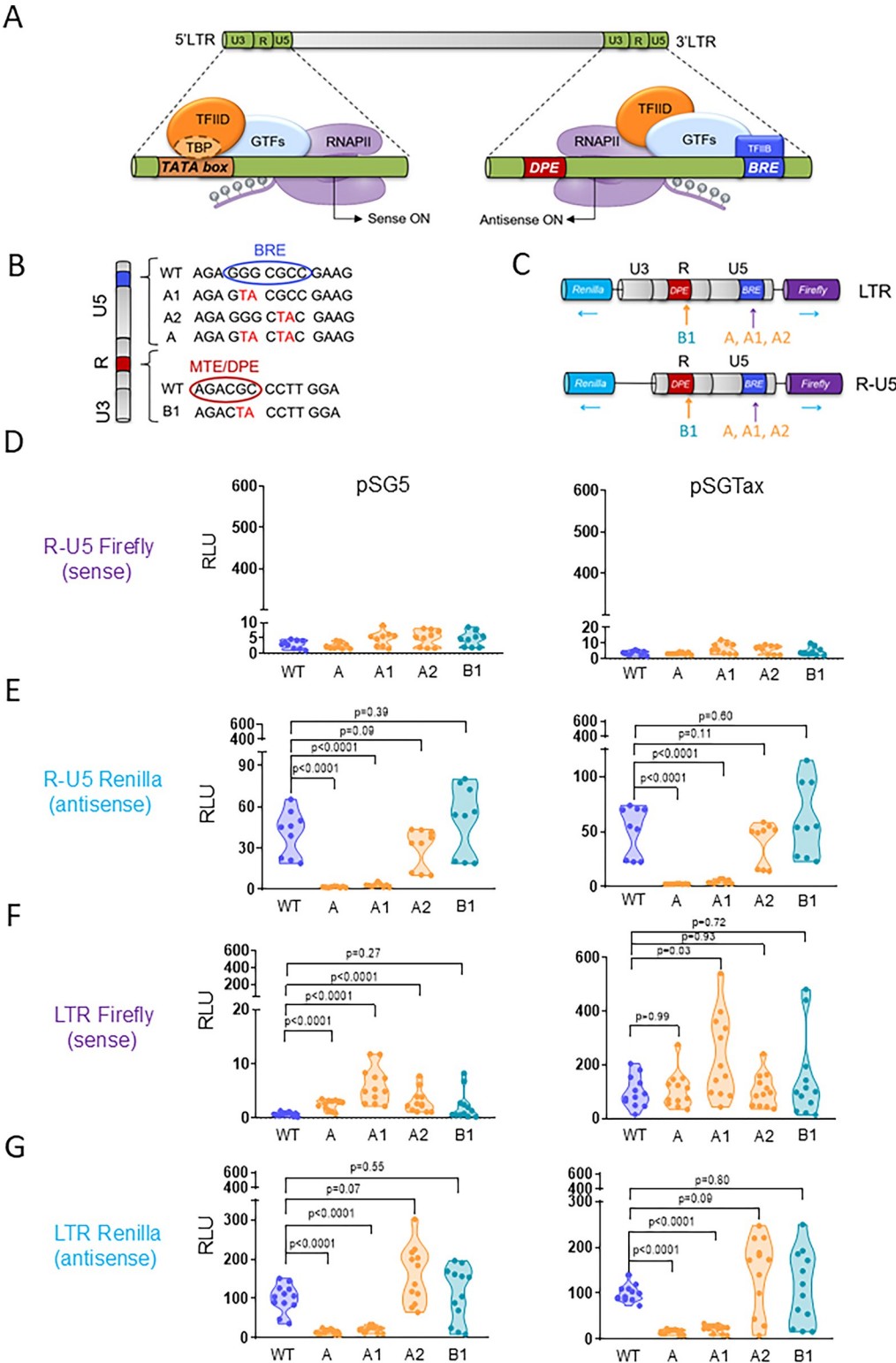

**Fig 1. The TFIIB recognition element is required for antisense transcriptional activity of the LTR promoter. (A)** Schematic representation of sense and antisense transcription initiation of the BLV provirus. Sense transcription requires the binding of the TATA-binding protein (TBP), a subunit of the transcription factor IID (TFIID), to the TATA box located in U3. TFIID further interacts with other general transcription factors (GTFs), resulting in recruitment of RNA polymerase type-II (RNAPII). Initiation of antisense transcription is predicted to be initiated by the

binding of TFIIB to the TFIIB recognition element (BRE) located in U5 **(B)** Nucleotide sequence of mutations (A, A1 and A2 in the BRE and B1 in the motif ten element/downstream promoter element MTE/DPE) introduced in the LTR and tested in reported assays. **(C)** Schematic representation of Firefly-Renilla dual-luciferase reporter plasmids containing either the full-length LTR or the R-U5 region. **(D-E)** Luciferase activities promoted by the WT and mutated minimal antisense promoter (R-U5). HEK293FT cells were co-transfected with the reporter plasmids (WT, A, A1, A2, B1) and a vector expressing either mock (pSG5) or the viral transactivator Tax (pSGTax). **(F-G)** Same experiment as described in panels D-E except that a dual reporter containing a full-length LTR was used. Twenty-four hours after transfection, Firefly and Renilla luciferase activities were measured and normalized to the corresponding mean luminescence generated by the full-length LTR reporter and pSGTax, arbitrarily set to 100. RLU: relative luminescence units. Data results from at least 3 independent experiments. p-values were calculated according to Mann-Whitney tests.

significantly reduced by the A1 mutation in absence and presence of Tax (**Fig 1G**). The A1 mutation also reduced antisense transcription in reporter assays performed with the Ovine Kidney (OVK) cell line (**S1 Data**).

These dual luciferase reporter assays thus revealed that a minimal two-nucleotide mutation (A1) in the BRE affects antisense transcriptional activity directed by the LTR promoter.

## A provirus carrying the A1 mutation transcribes less antisense RNAs, does not affect envelope-mediated cell fusion and is slightly more infectious in cell cultures

To evaluate the effect of antisense transcription in the proviral context, the A1 mutation was introduced in an infectious BLV molecular clone (pBLV344, **Fig 2A**). After transfection into HEK293FT, viral transcription, transmission to ovine cells and envelope-mediated cell fusion were analyzed by RT-qPCR, co-culture with OVK-U3Luc reporter cells and syncytia formation, respectively [27–29] (**Fig 2B**).

RT-qPCR quantification revealed that the levels of antisense RNAs were reduced in cells transfected with the A1 provirus compared to the WT control (p = 0.004; *as1* by 5-fold and *as2* by 11.9-fold; **Fig 2C**). In contrast, the WT and A1 viruses expressed similar amounts of *Tax* RNAs (p = 0.41), indicating that sense transcription remained mostly stable. To assess viral transmission, HEK293FT transfected with proviruses WT/A1 were then co-cultured with an ovine cell line transduced with a luciferase gene controlled by the U3 promoter (OVK-U3Luc). The luciferase activity increased by a factor of 1.7-fold (p = 0.03), suggesting that the A1 mutation may slightly improve viral infectivity (**Fig 2D**). However, the ability to promote cell fusion with CC81 indicator cells, as estimated by the number and the size of syncytia, was not statistically different (p = 0.55) (**Fig 2E** and **2F**). The coculture of HEK293FT with CC81 cells did not significantly modify the transcriptional pattern of the WT and A1 proviruses (**S2 Data**).

These observations thus show that, compared to the WT, the A1 provirus transcribes significantly less antisense RNAs, is slightly more infectious and induces similar levels of cell fusion events in transient transfection experiments.

## The A1 provirus is infectious in sheep but replicates at reduced proviral loads

To investigate the role of antisense transcription *in vivo*, sheep were inoculated with the WT proviral clone (n = 5) or the A1 mutant (n = 5). After 20 to 35 days, antibodies directed against the surface receptor-binding subunit of the viral envelope protein (gp51, SU) were detected in the plasma of all animals. The anti-gp51 reactivity was similar in sheep inoculated with the A1 and WT proviruses and persisted over time revealing the onset of an antiviral immune response (p = 0.64; **Fig 3A**). Quantitative PCR revealed that, compared to the WT, the proviral loads were reduced by 39-fold in the PBMCs of A1-infected sheep (p = 0.002; **Fig 3B**). The

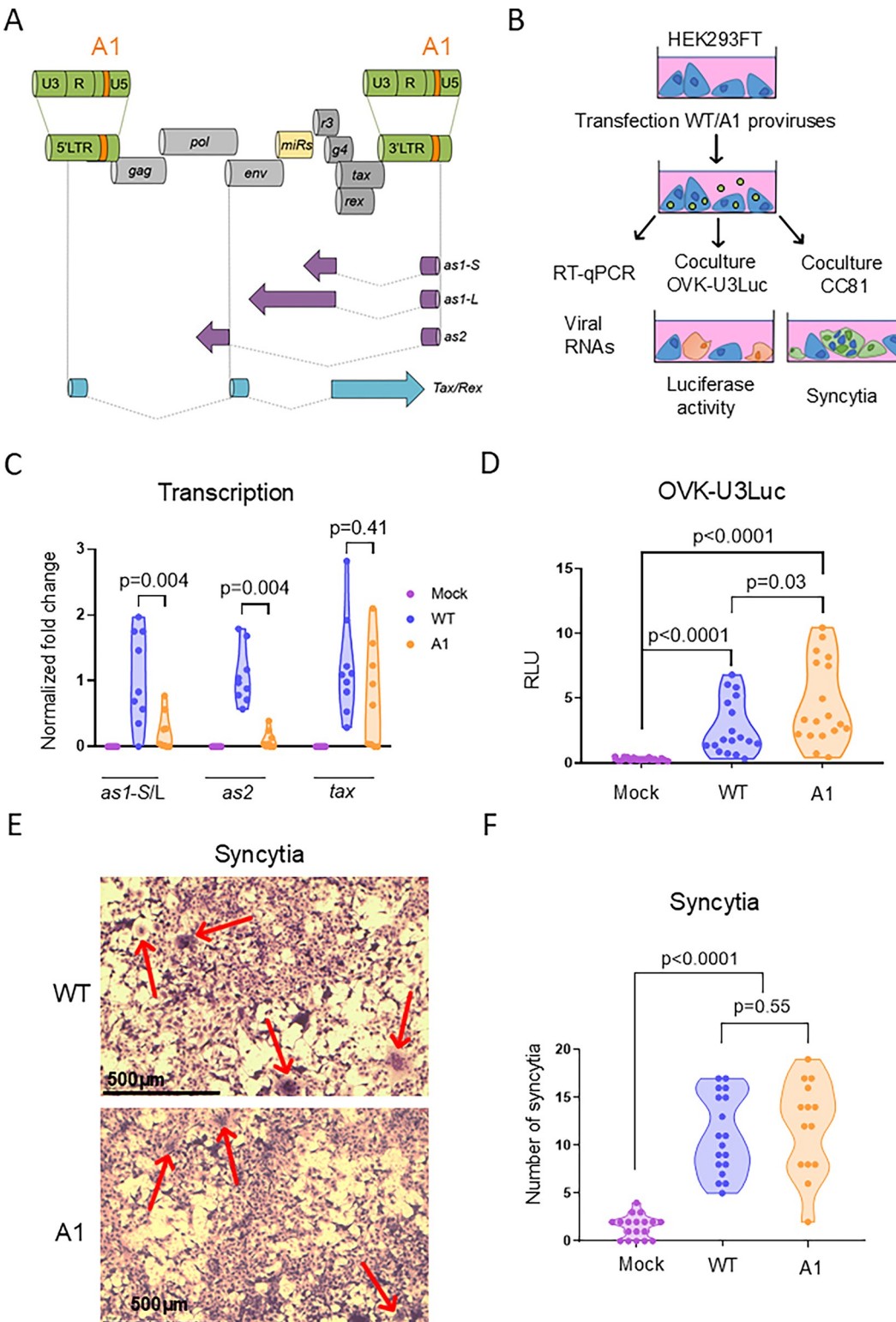

**Fig 2. Compared to the WT, a provirus containing the A1 mutation transcribes less antisense RNAs but is infectious and forms syncytia. (A)** Schematic representation of the provirus containing the A1 mutation (in orange): genomic structure, viral microRNAs (miRs in yellow), antisense transcripts (*as1-S*, *as1-L*, *as2* in purple) and the *Tax/Rex* mRNA (in blue). Note that other sense mRNAs encoding structural and regulatory proteins are not shown. **(B)** Experimental design. HEK293FT cells were transfected with either the WT provirus or the A1 mutant. At 48 hours post-transfection, HEK293FT

cells were collected to determine viral transcription, infection of OVK-U3Luc and syncytia formation with CC81 cells. **(C)** Quantification of *tax*, *as1-S/L* and *as2* mRNAs by RT-qPCR. Relative quantification was performed by the ΔΔCT method using ß-actin as reference [78]. Data resulting from three independent experiments were normalized to the mean of the WT samples, arbitrarily set to 1. p-values were calculated according to two-tailed Wilcoxon signed rank test **(D)** After 24 hours of coculture, luminescence activity was quantified in OVK-U3Luc reporter cells containing a stably integrated Firefly luciferase gene cloned downstream of the U3 promoter. Data in relative luminescence units (RLU) were normalized to the mean of the WT samples, arbitrarily set to 1. p-values were calculated according to Mann-Whitney tests. **(E)** Representative images of syncytia formed after five days of coculture between CC81 cells and HEK293FT transfected with WT/A1 proviruses. Cells were stained by May-Grünwald Giemsa and imaged with a Motic AE 2000 inverted light microscope. **(F)** The number of syncytia characterized by 10 or more nuclei in 9 images from 3 independent experiments were counted using the Motic 3.0 live software. p-values were calculated according to Mann-Whitney tests.

normalized RNA loads of *as1-S/L* and *as2* evaluated by RT-qPCR were also significantly decreased in the A1 PBMCs (p = 0.004 and p = 0.04 respectively; **Fig 3C**). In contrast, the levels of the *Tax* transcripts in the two groups were variable but not statistically different (p = 0.91). To evaluate the expression of viral proteins, PBMCs were transiently cultured overnight and analyzed by confocal microscopy using a monoclonal antibody directed against the p24 capsid protein (**Fig 3D**). At the single cell level, the p24 mean fluorescence intensities (MFIs) were similar in PBMCs infected with WT and A1 viruses (p = 0.42; **Fig 3E**).

To characterize the mechanisms ongoing in sheep infected with the WT provirus and the A1 mutant, the transcriptome of their B lymphocytes was determined by RNA sequencing. After isolation of B-cells by magnetic-activated cell sorting (MACS), RNAs were purified, converted to cDNA and sequenced with an Illumina NovaSeq6000 device (**Fig 3F**). After quality control and trimming, the sequencing reads (100bp, 30M, paired-end) were aligned on the *Ovis aries* reference genome (Oar_rambouillet_v1.0) using the HISAT2 (version 2.2.1) software. After alignment and quantification using featureCounts, differentially expressed genes were identified between cell populations using DESEq2 Bioconductor package. The volcano plot highlighted specific changes (DEG at p-adj < 0.05 and $|Log_2FC| > 1$) that correlated with infection with A1 and WT viruses (**S3–S6 Data**). Particularly, the *Hmcn1* gene, coding for the hemicentin-1 protein, was highly upregulated (p-adj = $5.29*10^{-127}$ and $|Log_2FC| = 7.46$; red arrow on **Fig 3G**). The heatmap of the 30 most significant DEG highlighted significant differences in the transcriptome of A1 and WT-infected B-cells (**Fig 3H**). To obtain a comprehensive view of the underlying mechanisms, an unsupervised analysis of the transcriptomic data set was performed. Gene Ontology (GO) comparison and pathway enrichment analysis revealed that KEGG pathways pertaining notably to proliferation (*oas04110* "Cell cycle", *oas04110* "DNA replication"), signal transduction (*oas04015*, "Rap1 signaling pathway", *oas04060* "cytokine-cytokine receptor interaction") and adhesion (*oas04514* "cell adhesion molecule", *oas04510* "focal adhesion") were the most significantly affected (**Fig 3I**). In contrast, the transcriptome was almost similar in the non-B cells from A1- and WT-infected sheep (**S3**, **S7–S9 Data**). Sashimi plots drawn by Integrative Genomics Viewer (IGV) confirmed that RNA loads were reduced in A1 PBMCs compared to WT but could not reveal any significant alteration of splicing events (**S10 Data**).

Altogether, these results reveal that the A1 mutant replicates at reduced proviral loads and deeply modifies the transcriptome of infected B-cells.

## Hemicentin-1 is upregulated in PBMCs and secondary lymphoid organs of A1-infected sheep

Among the DEGs identified by RNA sequencing, the expression of *Hmcn1* appeared to be particularly high in B-lymphocytes of A1-infected sheep compared to the WT and mock controls (NI) (**Fig 3**). To validate this observation independently, RT-qPCR showed that the

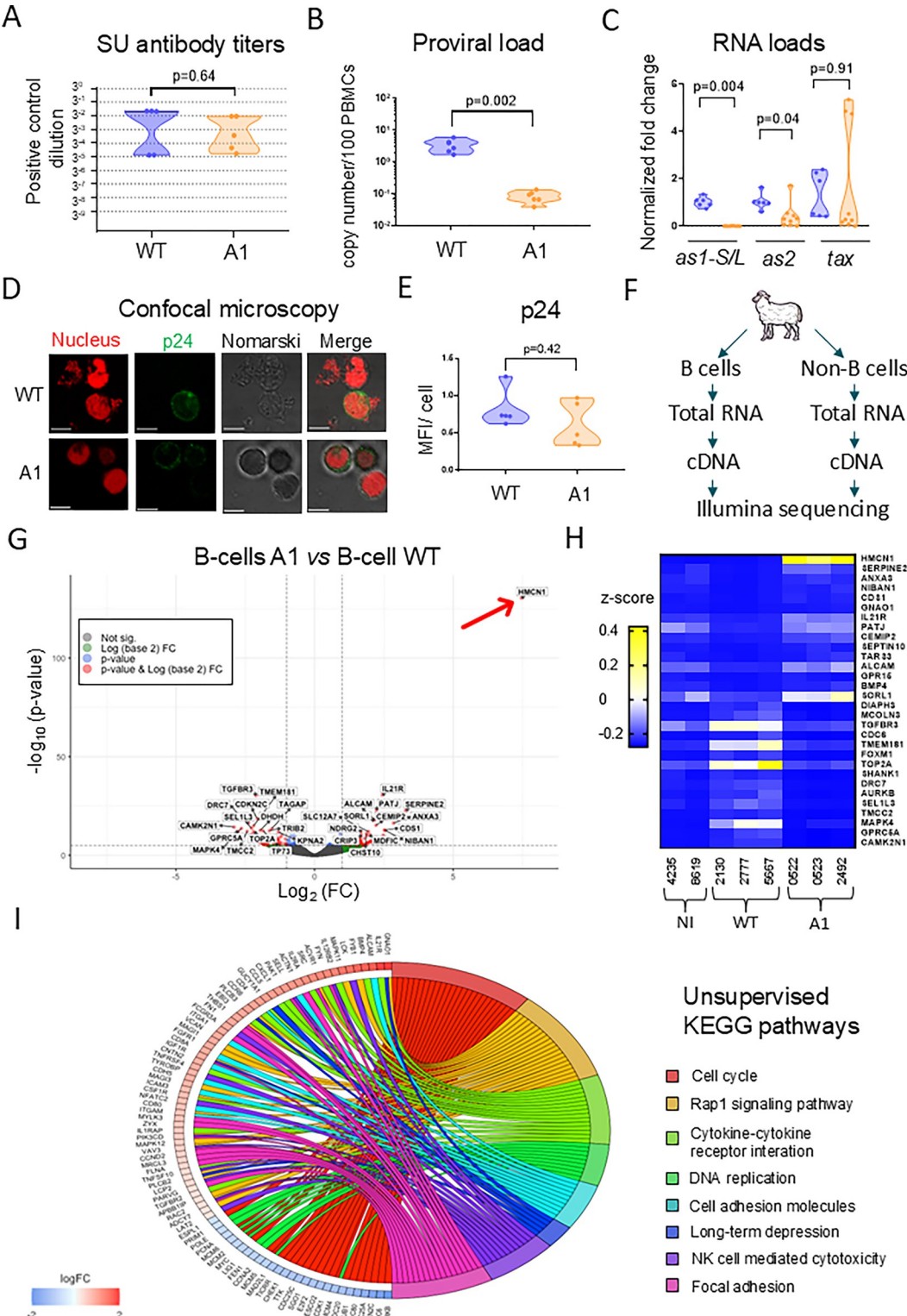

**Fig 3. The A1 mutant provirus is infectious in sheep and modifies the host cell transcriptome. (A)** Ten sheep were inoculated with either the wild-type provirus (n = 5) or the A1 mutant (n = 5). Anti-gp51 antibody titers were determined at 46 days post-inoculation using a blocking ELISA (IDEXX). The y axis represents three-fold serial dilutions of the plasma of a BLV-infected sheep with high proviral load (#3684) used as reference. **(B)** qPCR quantification of the proviral loads in sheep PBMCs. The y axis represents the number of proviral copies in genomic DNA (1 μg) of sheep #2130 (WT) and #0522 (A1).

The number of viral copies was normalized to a standard curve performed with the 18S cellular gene. p-values were calculated according to Mann-Whitney tests. **(C)** Quantification of *Tax*, *as1* and *as2* mRNAs by RT-qPCR. PMBCs were collected on day 811 post-inoculation. Relative quantification was performed by the ΔΔCT method using ß-actin as reference [78]. Data resulting from at least three independent experiments were normalized to the means of the WT sample, arbitrarily set to 1. p-values were calculated according to two-tailed Wilcoxon signed rank test. **(D)** Confocal microscopy images of PBMCs cultured overnight, stained with Draq5 (nuclei in red) and labelled with anti-p24 antibody (4'G9) combined with an AlexaFluor 488 conjugate (p24 in green). **(E)** Mean fluorescence intensity (MFI) of p24 labeling calculated using the ZEN Blue software (n = 5). p-values were calculated according to Mann-Whitney test. **(F)** RNA sequencing of PBMCs isolated from sheep infected with the WT or A1 proviruses. PBMCs (n = 3) were divided into B and non-B cell populations by MACS using the 1H4 anti-sIgM monoclonal antibody. After RNA extraction and reverse transcription, cDNAs were sequenced with an Illumina NovaSeq6000 device (100bp, paired-end, 30 million reads). **(G)** Volcano plot of most significant up- or downregulated genes in the B-lymphocytes of A1-infected sheep compared to WT controls. The volcano plot was generated using the EnhancedVolcano (version: 1.16.0) and ggplot2 (version 3.3.3) R packages. The x and y axes are $\log_2$(FC) and $-\log_{10}$(p-value), respectively. Cut-off values were defined as p-adj < 0.05 and $|\text{Log}_2\text{FC}| > 1$. The red arrow points to the *Hmcn1* gene. **(H)** Unsupervised heatmap of the 15 most significant (based on $\text{Log}_2$ FC) up- and down-regulated genes in B-cells from non-infected or A1- and WT-infected sheep. After PCA analysis, the non-infected sheep (#5751) was removed due to aberrant variation compared to samples of the same group. Aberrant behavior was further confirmed through correlation analysis. **(I)** Chord diagram of the most significant pathways affected in B-lymphocytes from A1-infected sheep compared to the WT. The left side of the graph represents the genes that are ranked by their $\log_2$(FC) and that are connected to the terms of the KEGG pathways on the right. Chord diagrams were generated using GOplot (version:1.0.2) R package.

normalized fold-change of *Hmcn1* RNA in A1-infected sheep exceeded that in NI and WT by 538 and 810-fold, respectively (p<0.0001; **Fig 4A**). Confocal microscopy further demonstrated that the expression of the hemicentin-1 protein encoded by the *Hmcn1* gene increased in A1-infected PBMCs and p24[+] cells compared to the WT (**Fig 4B**, **4C** and **4D**).

Hemicentin-1 (Fibulin-6 or ARMD1) is a member of the immunoglobulin superfamily that is expressed in a wide range of tissues including lymphoid conduits [30,31]. Hemicentin-1 mediates a wide range of mechanisms such as cytoskeleton rearrangements, mitosis [32,33], basement membrane invasion [30,34–36] and cell migration [37,38]. To investigate the role of hemicentin-1, OVK cells were transduced with lentivectors encoding *Hmcn1* shRNAs (sh1, sh2) or a scrambled (scrbl) control. Effective RNA interference of *Hmcn1* transcription was demonstrated by RT-qPCR (2.5-fold by sh1 and 3-fold by sh2; **Fig 4E**). The motility of shRNA-transduced OVK cells was then recorded for 48 hours by Incucyte live-cell imaging. Migration of individual cells was measured using the QuPath software and graphed using MatLab (**Fig 4F**). As hypothesized, knock-down of *Hmcn1* reduced the ability of OVK cells to undergo cell migration (p = 0.001 and p<0.001 for sh1 and sh2, respectively; **Fig 4G**).

To extend these observations, the expression of hemicentin-1 was determined by immuno-histochemistry in lymphoid organs from WT- and A1-infected sheep. Hemicentin-1 was expressed in the spleen and lymph node of an A1-infected animal (**Fig 4H and 4I**). In contrast, hemicentin-1 could not be detected in a biopsy of a prescapular lymph node isolated from a WT-infected sheep (**Fig 4I**). Expression of hemicentin-1 was significantly higher in the spleen of A1-infected sheep compared to the wild-type (n = 7.21 and n = 1.68 hemicentin-1[+] cells / mm[2] in A1 and WT biopsies, respectively). Confocal microscopy revealed that hemicentin-1 predominantly aggregated in the intercellular space within lymphoid organs (**Fig 4J** and **S11 Data**).

Extending the RNA sequencing data (**Fig 3**), it appears that a factor regulating cell migration, the hemicentin-1 protein, is overexpressed in PBMCs and in lymphoid organs from sheep infected by the A1 mutant.

## The A1 mutation attenuates oncogenesis in sheep

The pathogenesis induced by BLV is tightly controlled by the immune response mediated by lymphoid organs. In particular, cell proliferation in the spleen has a central role in virus-driven

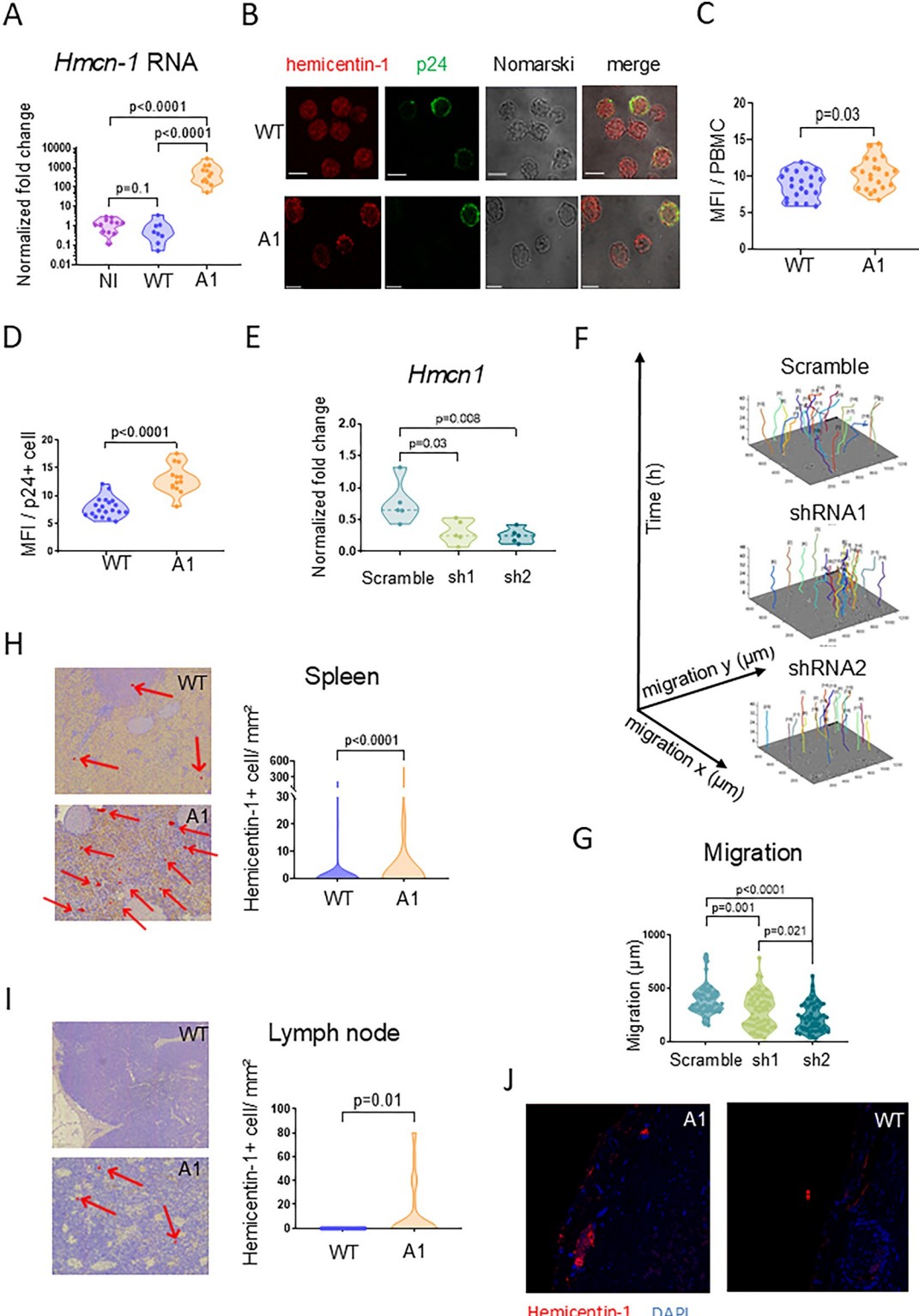

**Fig 4. HMCN-1/hemicentin-1 is upregulated in peripheral B-cells, lymph nodes and spleen of A1-infected sheep. (A)** RT-qPCR quantification of *Hmcn-1* RNA in PBMCs collected at day 811 post-injection. Relative quantification was performed by the ΔΔCT method using ß-actin as reference [78]. Data resulting from three independent experiments were normalized to the means of WT samples, arbitrarily set to 1. p-values were calculated according to the two-tailed Wilcoxon signed rank test. **(B)** Expression of hemicentin-1 protein in *ex-vivo* cultures of BLV-infected PBMCs. Hemicentin-1 was labelled with the

HPA051677 rabbit antiserum and an AlexaFluor 647 secondary antibody (in red). The p24 capsid protein was revealed by the 4'G9 monoclonal antibody and an AlexaFluor 488 conjugate (p24 in green). Scale bars indicate 5 μm. **(C)** Quantification of the mean fluorescence intensity (MFI) of hemicentin-1 in PBMCs from sheep #2130 (WT) and #0522 (A1) measured using the ZEN Blue software (n = 30 cells). **(D)** MFI of hemicentin-1 in cells expressing the p24 protein. **(E)** The OVK cell line was transduced with lentivectors encoding two *Hmcn-1* shRNAs (sh1, sh2) or a scrambled control. *Hmcn-1* RNA levels in transduced cell lines were quantified by RT-qPCR. Relative quantification was performed by the ΔΔCT method using ß-actin as reference [78]. Data resulting from five independent experiments were normalized to mock OVK cells, arbitrarily set to 1. p-values were calculated according to the Mann-Whitney test. **(F)** OVK cells transduced with lentivectors encoding scrambled/sh1/sh2 shRNAs were recorded for 48 hours using the Incucyte live-cell imaging system. Tracking plots of cell migration over time (in μm) of 20 individual cells identified manually by the Celltracker and modelized with the Matlab software. **(G)** Migration distances covered by 60 individual cells during the culture as measured with Celltracker. **(H)** Representative immunohistochemistry of hemicentin-1 in spleen biopsies collected from sheep #2130 (WT) and #0522 (A1). The numbers of hemicentin-1+ cells per mm$^2$ of spleen tissue. p-values were calculated according to Mann-Whitney tests. **(I)** Hemicentin-1 immunohistochemistry in lymph node biopsies isolated from sheep #2130 (WT) and #0522 (A1). Enumeration of hemicentin-1+ cells per mm$^2$ of spleen tissue. p-values were calculated according to Mann-Whitney tests. **(J)** Spleen sections from WT and A1-infected sheep were stained with DAPI (nuclei in blue) and labelled with anti-HMCN1 antibody combined with an AlexaFluor 647 conjugate (in red). Slides were imaged with a Stellaris confocal microscope (Leica).

leukemogenesis [39,40]. To evaluate proliferation, Ki67-based immunochemistry was performed on biopsies of lymphoid organs. Compared to WT, the number of Ki67$^+$ cells was moderately reduced by 1.73-fold in the lymph nodes of sheep infected with the A1 mutant (p<0.0001; **Fig 5A**). In contrast, cell proliferation in the A1 spleen increased by a factor of 9 compared to WT (p<0.0001; **Fig 5B**). The number of proviral copies per μg of genomic DNA determined by qPCR dropped by 54.3- and 197.8-fold in the lymph node and the spleen of A1-infected sheep (p = 0.002; **Fig 5C and 5D**). When normalized to the total viral load, the spatial distribution of the A1 proviruses in the PBMCs and the lymphoid organs was affected. Compared to WT, the relative proportion of A1 proviral DNA was reduced by 4.3-fold in the spleen (**Fig 5E**). In contrast, the percentages of A1 and WT proviral DNA were proportionally similar in the lymph nodes (1.9% and 1.2%, respectively). In other terms, attenuated replication of the A1 mutant correlated with increased cell proliferation and reduced spatial distribution of the proviruses in the spleen.

The five sheep inoculated with either the WT provirus or the A1 mutant remained persistently infected as indicated by the continuous presence of plasmatic antiviral antibodies (**Fig 5F**). The PVLs were significantly lower in the A1-infected sheep than in the WTs (p = 0.0001; **Fig 5G**). While the WT provirus was pathogenic, none of the A1-infected sheep developed any clinical manifestation of leukemia or lymphoma (**S12 Data**). Quantitative PCR and sequencing revealed that the two-nucleotide mutation introduced in the BRE of the A1 proviruses did not revert to WT (illustrated for #0582 in **Fig 5H**).

Collectively, these data thus show that the A1 mutant replicates at reduced proviral loads and attenuates pathogenicity in sheep.

## Discussion

The main take home message of this study is that antisense transcription is involved in BLV persistence and pathogenesis in sheep. The outcomes of this fundamental research for management of bovine leukemia/lymphoma are unclear but may be informative to better understand the mechanisms associated with the pathogenesis of HTLV-1. Indeed, the relevance of the animal models developed to characterize HTLV-1-associated diseases is sometimes questionable. Transgenic mice have been useful to demonstrate the oncogenic potency of viral genes such as Tax and HBZ [41,42]. Since these models are independent of viral replication, rabbits and humanized mice may provide a better overview but fail to recapitulate a fully competent immune response [43]. The same questions addressed in different models may also generate contradictory conclusions [44]. Among primates, cynomolgous and rhesus macaques

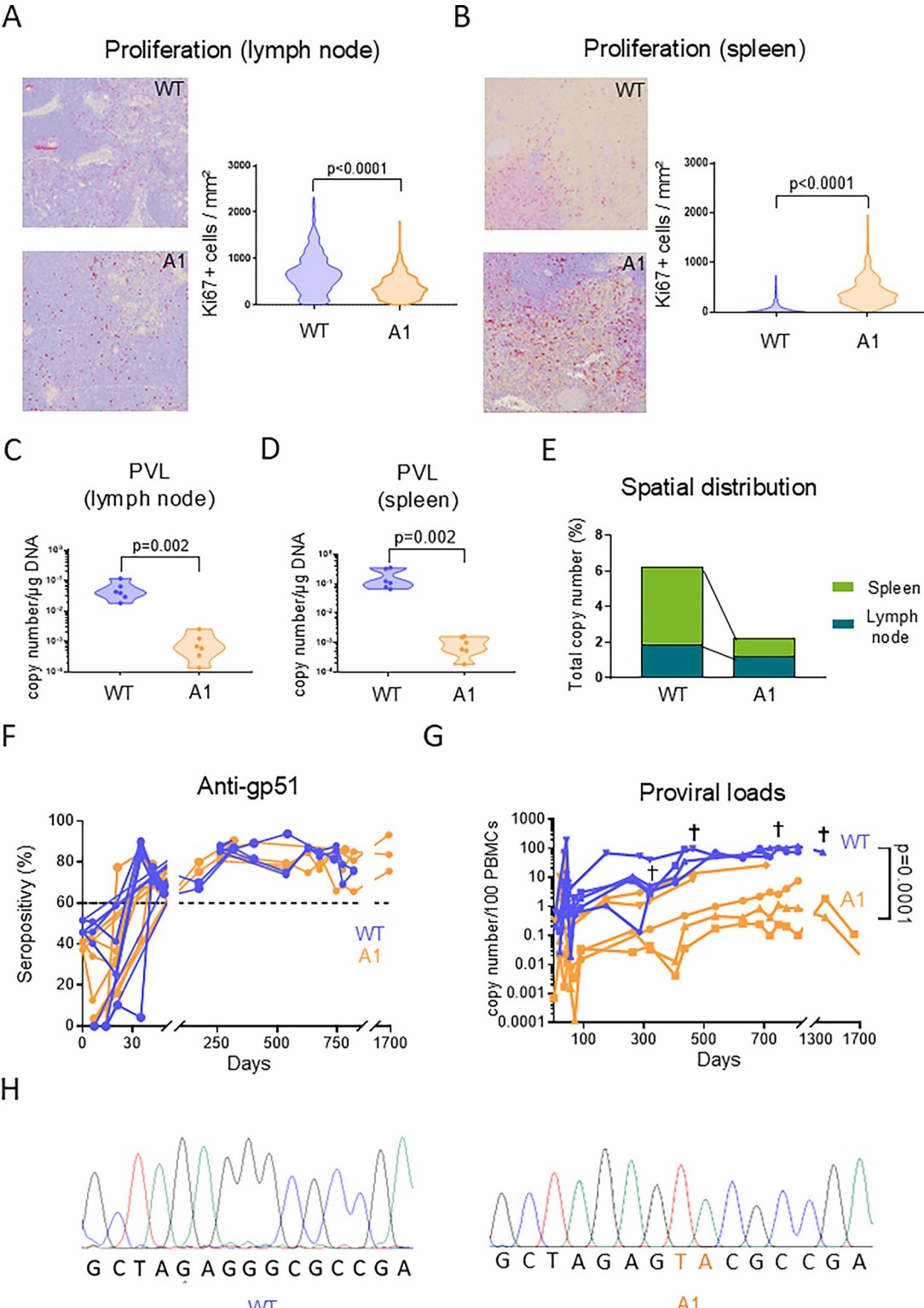

**Fig 5. The A1 provirus affects cell proliferation in lymphoid tissues and has reduced pathogenicity in sheep. (A)** Immunohistochemistry of Ki67 labelling (in red) in the prescapular lymph node. The numbers of Ki67+ cells per mm$^2$ were quantified using the QuPath 0.3.0 software. p-values were calculated according to Mann-Whitney tests. **(B)** Same experiment with spleen biopsies as described in panel A. **(C)** Quantification of the proviral loads in the prescapular lymph nodes. The number of proviral copies in genomic DNA (1 μg) of sheep #2130 (WT) and #0522 (A1) was measured by qPCR. The number of

viral copies was normalized to a standard curve performed with the 18S cellular gene. p-values were calculated according to Mann-Whitney tests. **(D)** Same experiment as described in panel C performed with spleen biopsies. **(E)** Comparison of the viral spatial distribution between the lymph node and the spleen of WT- and A1-infected sheep. **(F)** At regular time intervals, plasma was collected and tested for the presence of anti-gp51 antibodies using a blocking ELISA (IDEXX). The dashed line represents the seropositivity threshold. **(G)** The proviral loads in PBMCs of infected sheep were quantified by qPCR. The y axis represents the number of viral copies per 100 PBMCs. p-values were calculated according to Wilcoxon matched-pairs signed rank test. Note that data result from 2 independent experiments. † means dead. **(H)** Genomic DNA was extracted from PBMCs of WT- or A1-infected sheep (#5667 and #0582, respectively). Representative sequencing profiles of the LTR sequence amplified by PCR.

[45,46] have been very useful models but are inadequate to study HTLV-1 associated pathogenesis because of the low frequency of leukemia and the long latency period. To complement the conclusions drawn from these different models, a comparative virology approach of BLV-infected sheep may thus broaden the understanding of retroviral-induced oncogenesis.

In the present report, the relevance of antisense transcription has been tackled by a reverse genetics approach. This strategy has been useful to highlight the relevance of genetic determinants involved in infectivity, replication and oncogenicity [25,28,47]. The intrinsic drawback of reverse genetics is that a targeted mutation may affect several functional elements because of the complexity of a condensed and relatively small BLV genome (~ 8.7 Kbp) [48,49]. In particular, a major difficulty of addressing antisense transcription is that mutations in one strand would also have affected the other. Our strategy to investigate the relevance of antisense transcription has been based on mutations introduced in the LTRs flanking the BLV genome. Although the two-nucleotide mutation in the BRE is limited, it cannot be excluded that other mechanisms are also affected. What is worthy of note, however, is that the mutation in the BRE does not overlap the rex-responsive element that mediates post-transcriptional regulation of viral gene expression [50,51]. Since the process of reverse transcription duplicates the two terminal sequences, a mutation of the 3' antisense promoter is copied into the 5'-LTR promoter. In reporter assays using a single LTR flanked by Firefly and Renilla luciferase genes, sense transcription appears to be increased by the A1 mutation in HEK293FT but not in OVK cells (**Fig 1F** and **S1 Data**), revealing cell type specificities. Since the increase of sense transcription was observed neither in the context of a full-length provirus (**Fig 2C**) nor *in vivo* (**Fig 3A–3E**), it appears that the A1 molecular clone optimally fulfils the different characteristics required to address the role of antisense transcription.

Why is replication of the A1 mutant affected in sheep? Among hypotheses, the collisions between RNA polymerases transcribing opposite DNA strands may be reduced and favor the expression of structural/enzymatic genes thereby exposing the virus to the host immunity. Similarly, mutations of the Tax-response elements that optimize the CRE elements increase promoter activity but interfere with viral replication in sheep [25]. As indicated in the former paragraph, we think that this paradigm is unlikely because (1) transcription of *tax* RNA from A1 and WT proviruses is similar (**Fig 2C**), (2) there is only a modest increment in infectivity (**Fig 2D**), (3) there is no significant difference in cell fusion (**Fig 2F**) and (4) the expression of p24 protein is identical at the single cell level in vivo (**Fig 3E**). Since sense transcription is mostly unaffected, it is also unlikely that the A1 mutation reduces steric hindrance of the polymerase complex at the 5'-LTR. Notwithstanding that the A1 mutation does not overlap the polypurine tract (PPT) and the primer binding site (PBS) (**S13 Data**), it is predicted that the process of reverse transcription of the A1 provirus is also similar to WT. Other hypotheses that may explain the reduced replication of A1 more directly involve an epigenetic role of the antisense RNA as demonstrated for HTLV-1 [23], the activity of a putative small peptide encoded by 264bp open reading frame (ORF) [52] or cis-perturbation of cancer driver genes in the host genome [53].

Whatever the molecular mechanism, it appeared that the expression of *Hmcn1*/hemicentin-1 was particularly high in A1-infected cells as shown by RNA sequencing (**Fig 3**), RT-qPCR (**Fig 4A**), confocal microscopy of PBMCs (**Fig 4B–4D**), and immunochemistry of lymphoid tissues (**Fig 4H–4J**). As mentioned above, antisense RNAs may modulate *Hmcn1* expression by epigenetic modifications [23], or alternatively via transcription factors such as cFos known to mediate HMCN1 promoter activity [54,55]. However, luciferase-based reporter assays with a plasmid controlled by sequences overlapping the *Hmcn1* promoter did not reveal significant difference after transfection of A1 and WT proviruses (**S14 Data**). suggesting an indirect mechanism ongoing inside infected cells. Similarly, it is unclear why *Hmcn1* expression is also increased to a lesser extent in the non-B cell population (**S3 and S7 Data**). Evidence indicating that *Hmcn1*/hemicentin-1 can be indirectly induced by cytokines (TGFβ), glucose and signaling pathways (MAPK) [55,56] needs to be specifically addressed by further experiments.

Although the role of hemicentin-1 in BLV persistence and replication is currently unknown, RNA interference shows that cell migration is affected (**Fig 4E–4G**). This phenotype is consistent with the involvement of hemicentin-1 in TGFβ-mediated cytoskeleton rearrangements, organization of F-actin fibers [55] and architecture of epithelial cell junctions. Importantly, hemicentin-1 also promotes cleavage furrow maturation during cytokinesis [33,35]. The expression of hemicentin-1 detected by immunohistochemistry (**Fig 4H–4J**) may thus indirectly reflect cell mitosis and proliferation, as previously observed in B and non-B cells infected with a provirus lacking the miRNAs (see S1 Table of reference [57]). It thus appears that alterations of hemicentin-1 expression likely affect important mechanisms such as cell migration or mitosis. Therefore, an overall hypothesis may be summed up as follows: through an indirect mechanism, the lack of antisense RNAs increases the production of hemicentin-1 protein, which accumulates mostly as tracks in the extracellular space of lymphoid organs. These tracks may affect leukocyte trafficking ultimately resulting in a better control of the proviral load and oncogenesis.

Addressing antisense transcription by reverse genetics in the ovine model reveals an essential role exerted by the lymphoid organs. This experimental evidence extends previous observations showing that splenectomy accelerates pathogenesis in sheep suggesting that the spleen is required for antiviral immunity [58]. Homeostasis is preserved providing an equilibrium between cell proliferation and death in lymphoid organs and peripheral blood. For example, the deletion of the viral microRNAs from the BLV genome (*i.e.*, mutant ΔmiRNA [57]) leads to a reduction of proliferation both in lymph nodes as well as in spleen. In vivo kinetic studies based on CFSE and BrdU labeling revealed a concomitant reduced proliferation in the peripheral blood of sheep infected with the ΔmiRNA deletant. Cell dynamics that characterize the A1 mutant are different because proliferation is reduced in the lymph node but increased in the spleen (**Fig 5**). It thus appears that unevenness associated with the kinetics of the ΔmiRNA and A1 mutants leads to attenuation of their pathogenic activity.

Because the A1 proviral loads in the lymphoid organs is extremely low (**Fig 5C and 5D**), the high rates of cell proliferation revealed by the KI67 labelling (**Fig 5A and 5B**) mostly correspond to uninfected cells. It is thus likely that these cells are T-lymphocytes that are activated by antiviral immunity. As supported by many previous reports, BLV infection is associated with an increase in T cell proliferation as a marker of an active antiviral immune response [59,60]. Experiments based on intravenous injection of CFSE and BrdU would precisely quantify the cell kinetics in vivo [40,57,58]. In BLV-infected sheep, antiviral immunity is centrally controlled by the spleen because splenectomy accelerates the onset of leukemia/lymphoma [58]. On the other side, provirus-carrying B cells undergo active proliferation to persist, accumulate and ultimately induce leukemia in sheep [61]. Similarly, the HTLV-1 induced Adult T-cell leukemia/lymphoma (ATLL) and HTLV-associated myelopathy / tropical spastic

paraparesis (HAM/TSP) are also characterized by active proliferation of provirus-carrying cells [62]. This interpretation is consistent with the mostly accepted paradigm of a virus that attempts to replicate in presence of a vigorous antiviral immunity. This model postulates that the virus undergoes cycles of transcriptional silencing and transient burst of expression allowing infection of new cells. Challenging these cycles with epigenetic activators leads to a reduction of the proviral loads and has therapeutic value in BLV-infected sheep [12,63,64].

Does the present report open perspectives for therapy of BLV-induced leukemia/lymphoma? Currently, management of EBL aims at preventing viral prevalence in herds by genetic selection based on specific DRB3 alleles [65,66], selective eradication of animals with high PVL [9] and vaccination [11]. Because of economical constraints, therapy of EBL is not considered and, therefore, interfering with antisense transcription of BLV is not an adequate option. For the related HTLV-1 however, the use of shRNA interference or peptide nucleic acids targeting *hbz* may be useful to reduce viral replication. Alternatively, disequilibrate the balance between sense and antisense transcription for example with pharmacological inhibitors, epigenetic modifiers and perhaps hemicentin-1 inducers may have therapeutic value [46,67–70]. Clinical trials are required to address the effectiveness and safety of these different approaches against HTLV-1.

## Materials and methods

### Ethics statement

Animal experimentation was conducted in accordance with national (Royal Decree on the Protection of Experimental Animals) and international (European Union) guidelines. Sheep were kept under L2-restricted conditions (registration number LA1900600), according to the ethical protocols n˚1515/2067, approved by the *Commission d'éthique de l'utilisation des animaux à l'université de Liège* (Uliège), at the *Centre d'Etude en Production Animales* (CEPA) facility located in Gembloux (Belgium).

### Plasmid constructions

The full-length LTR (531bp) and the R-U5 fragment (last 132bp) of the pBLV344 provirus [71] were inserted into the KpnI and PstI restriction sites of the dual luciferase reporter plasmid pAsLuc(Fire)-HTLV-Luc(Reni) [72] using standard molecular biology techniques. In these pAsLuc(Fire)-531bp-Luc(Reni) and pAsLuc(Fire)-132bp-Luc(Reni) constructs, the HTLV LTR was thus replaced by BLV sequences. Mutations (A, A1, A2 and B1) were introduced in these reporter vectors using the XL QuickChange II Site-Directed Mutagenesis kit (Agilent Technologies) and primers listed in **S15 Data.** Mutation A1 was also introduced in both LTRs of plasmid pBLV344 containing a full-length wild-type provirus [71], yielding pBLV344-A1.

The pU3BLV-Luc lentivector (Vector ID VB201221-1032xzh; VectorBuilder) contains the 79bp BLV enhancer element [73], cloned upstream of the pGL3Basic minimal promoter and the humanized Luc2 Firefly luciferase gene. Lentivectors coding for short hairpin RNA against *Hmcn1* (shRNA1 5'-AGCGATAAAGGGACATATATT-3' and shRNA2 5'-CAAAGATGGC AAGCCTTTATT-3') were obtained from VectorBuilder (Vector ID VB220127-1038fqz and vector ID VB220127-1040mmq, respectively). The scrambled control (GVV_659 pLV shRNA NT) was provided by Emmanuel Di Valentin (GIGA viral vector platform). The packaging plasmids pSPAX2 (#12260) and VSV.G (#14888) were obtained from Addgene.

### Cell culture and lentiviral transduction

Human embryonic kidney cells (HEK293FT; Cellosaurus CVCL_6911) and OVK [28] were cultured in Dulbecco's modified Eagle's medium (DMEM, Biowest) supplemented with 10%

decomplemented fetal bovine serum (FBS, Gibco) and 1% of penicillin-streptomycin solution 100x (VWR). The CC81 feline cell line [29] (Cellosaurus CVCL_7216) was cultured in Roswell Park Memorial Institute 1640 medium (RPMI 1640, Biowest) supplemented with 10% decomplemented FBS and 1% of penicillin-streptomycin solution. All cultures were maintained at 37˚C, in a humidified 95–5% air/$CO_2$ incubator (Memmert ICO150).

The OVK-U3Luc, OVK-shRNA1, OVK-shRNA2 and OVK-Scramble cell lines were generated by lentiviral transduction of the OVK cell line. Lentivectors (429 ng) encoding scrambled/sh1/sh2 shRNAs and pU3BLV-Luc were co-transfected in HEK293FT ($2.5x10^5$ cells/well of a 6-well plate) with VSV.G (857 ng) and pSPAX2 (1.714 μg) using Jetprime (Westburg). Supernatant (2 mL) of transfected HEK293FT cells was harvested 72 hours post-transfection, filtered through 0.45 μm cellulose membrane filters and added to OVK ($5x10^5$) in presence of hexadimethrine bromide (3.66 μg/mL; Sigma-Aldrich). At 72 hours post-transduction, antibiotic selection was performed for two weeks with blasticidin (20 μg/mL; InvivoGen).

## Dual-luciferase assays

The WT and mutated bidirectional reporters (50 ng pAsLuc(Fire)-531bp-Luc(Reni) and pAsLuc(Fire)-132bp-Luc(Reni) were cotransfected with 300 ng of a plasmid encoding Tax (pSGTax [26]) or a control (pSG5) into HEK293FT cells ($5x10^5$ cells/well of a 6-well plate) using Jetprime reagent. Cells were lysed at 24 hours post-transfection and Renilla/Firefly luciferase activities were measured with a luminometer (Turner Designs TD-20/20) using the Dual-Luciferase Reporter Assay system (Promega), according to the manufacturer's protocol.

## Syncytia formation

The pBLV344 and pBLV344-A1 plasmids (20 μg) were mixed with 21 μg of polyethyleneimine (25 kD, Polysciences) in OPTI-Minimum Essential Medium (OPTI-MEM, Gibco) and added to HEK293FT cells ($2x10^6$ cells in a 10 cm diameter culture plate) for 24 hours. After replacement of the medium, HEK293FT cells were cultured for 24 hours, washed with Dulbecco's Phosphate Buffered Saline (DPBS, Biowest) and detached with trypsin-EDTA (Biowest). Transfected HEK293FT ($3x10^5$) cells were added to CC81 cells ($1.5x10^6$) in a 10 cm diameter culture plate. After 72 hours, cells were washed and stained in 5mL May-Grünwald buffer (Sigma-Aldrich) for 3 minutes at room temperature. After addition of 5mL of water for 1 minute, the stain was removed and replaced by 10 mL of 10% giemsa's azue methylene blue solution (Merck) for 20 minutes. Finally, the stain was removed, and cells were washed with 10 mL of neutral water.

The number of syncytia with more than 10 nuclei were scored by two different operators using a inverted light microscope (Motic AE 2000). Representative pictures were taken with the Motic 3.0 live software.

## In vitro infectivity assays

Plasmids pBLV344 and pBLV344-A1 (1 μg) were transfected using Jetprime into HEK293FT cells ($5x10^5$ cells/well of a 6-well plate). After replacement of the medium 24 hours post-transfection, cells were cultured for one day and mixed with OVK-U3Luc ($5x10^5$ cells/well of a 6-well plate). 24 hours later, Renilla and Firefly luciferase activities were measured with a luminometer using the Dual-Luciferase Reporter Assay system, according to the manufacturer's protocol.

## Live cell imaging

OVK-scramble, OVK-shRNA1 and OVK-shRNA2 cells ($9x10^3$ cells/well of a 96-well plate) were cultured at 37˚C in a humidified 95/5% air/$CO_2$ incubator of an Incucyte imaging S3 Live-cell system (Sartorius). Migration of individual cells was recorded every 15 minutes using the CellTracker software (v1.1) [74]. Each frame underwent vignetting correction by bicubic interpolation and frames were aligned prior to the analysis. Manual tracking of each individual cell was performed, with a linear interpolation between each frame. Motility graphs were generated using the MATLAB software (MathWorks).

## Inoculation of proviruses into sheep

Plasmids pBLV344 or pBLV344-A1 (20 μg) were transfected into HEK293FT cells ($9x10^6$ cells in a 15 cm diameter culture plate), using 60 μL of polyethyleneimine (1 μg/μL, 25 kD, Polysciences) in 1 mL of OPTI-MEM. After overnight culture, cells were washed with DPBS and cultured in fresh medium for 24 hours. Transfected HEK293FT cells were detached with trypsin-EDTA, pelleted by centrifugation and resuspended in 1 mL DPBS. After a freeze-thaw cycle at -80˚C, the cell lysate was inoculated intravenously and subcutaneously into sheep.

At regular time intervals, whole blood was collected by jugular venipuncture using 2% ethylenediaminetetraacetic acid (v/v) (EDTA, VWR) as anticlotting agent. After collecting the plasma, PBMCs were separated by Percoll (Merck) gradient centrifugation and preserved at -80˚C in FBS containing 10% DMSO. The detection of anti-gp51 antibodies in the plasma was performed by ELISA (BLV gp51 Antibody Test Kit; IDEXX), according to the manufacturer's protocol. Optical density at 450nm ($OD_{450}$) was measured with a Multiskan GO Microplate Spectrophotometer (Thermo Fisher Scientific). The seropositivity score was calculated using the following formula: 100–100 multiplied by ($OD_{450}$ of the sample to be tested divided by $OD_{450}$ of the negative control). The sample was considered seropositive providing that the score was > 60%.

## Proviral load quantification by qPCR

Genomic DNA was extracted from PBMCs using the Blood and Tissue kit (Qiagen), according to the manufacturer's protocol. The PCR mix contained 2.5 μL of DNA at 20 ng/μL, 6.25 μL of TAKYON ROX master mix (Eurogentec), 0.25 μL of forward and reverse primers at 20 μM (**S15 Data**) and 3.25 μL of milliQ-grade water. Real-time PCR amplification was performed in a thermocycler (Stepone Real-Time PCR, Applied Biosystems). The thermal protocol was initiated by a 10 minutes denaturation step at 95˚C, followed by 40 cycles (15 sec at 95˚C, 60 sec at 60˚C) and terminated by a melting curve (1 minute at 60˚C, increasing 0.3˚C steps every 15 sec up to 95˚C). The number of viral copies was normalized to a standard curve performed with the 18S cellular gene (2*BLV/18S), as described previously [28,75]. The number of viral copies per cell was then obtained by multiplying the BLV/18S ratio by the average value of the number of 18S copies per cell. The spatial distribution of the virus was determined by comparing the calculated proviral copy number measured in 1μg of DNA extracted from PBMCs, spleen and lymph node to the total amount of proviral copies detected in these tissues.

## RNA quantification by RT-qPCR

RNA was extracted from transfected cells and PBMCs using the NucleoSpin RNA kit (Macherey-Nagel) according to the manufacturer's protocol. Total RNA (200 ng) was converted to cDNA with the Reverse Transcriptase core kit (Eurogentec). 2 μL of complementary DNA (cDNA) was amplified by qPCR, using the same thermocycling protocol as described

above, using primers corresponding to *hmcn1*, actin and viral transcripts (*as1*, *as2* and *Tax*; **S15 Data**). Relative quantification was performed using actin as housekeeping gene.

## Sequencing of proviral amplicons

DNA was extracted from PBMCs using the DNeasy Blood and Tissue kit (Qiagen), according to the manufacturer's protocol. Genomic DNA (500 ng) was amplified in a mix containing 1 μL forward and reverse primers (20 μM round 1; **S15 Data**), 0.25 μL Taq polymerase (500 U/μL, Thermo Scientific), 5 μL Taq buffer 10X (Thermo Scientific), 3 μL $MgCl_2$ (25mM, Thermo Scientific), 1 μL dNTP mix (2 mM, Thermo Scientific) and 36.75 μL miliQ-grade water. The protocol of the thermocycler (Veriti 96-Well Thermal Cycler, Applied Biosystems) was initiated by a 5 minutes denaturation step at 95˚C, followed by 40 cycles (30 sec at 95˚C, 30 sec at 67˚C, 1 minutes at 68˚C) and terminated by 5 minutes at 68˚C. A 2 μL aliquot of the PCR mix was reamplified with round 2 primers at an annealing temperature of 58˚C. DNA amplicons were migrated onto a 1% agarose gel, purified using the QIAquick gel extraction kit (Qiagen) and sequenced by the GIGA genotranscriptomic platform. Chromatograms were analyzed using the Chromas software (Technelysium).

## Immunohistochemistry of lymphoid organs

Spleen and lymph node biopsies were fixed in 4% paraformaldehyde (PFA, Sigma-Aldrich) for 24 hours and conserved in 70% ethanol (VWR). Biopsies were then embedded in paraffin, sectioned with a motorized microtome (Microm HM355S, Thermofischer Scientific / RM2165, Leica) and deposited onto glass slides. Deparaffinization and rehydration were performed by successive washes: xylene (VWR) (n = 3; 5 minutes), 100% ethanol (n = 2; 10 minutes), 95% ethanol (n = 2; 10 minutes) and distilled water (n = 2; 5 minutes). Antigen was retrieved in a pressure cooker (Ultra Cell Conditioning Solution, 950–224, Roche) in either EDTA (11 minutes, Invitrogen) or citrate buffer (15 minutes, Invitrogen). After cooling for 30 minutes, slides were washed three times with Tris-buffered saline with Tween 20 (TBST) (pH 7.4) under gentle agitation and incubated one hour at room temperature with a blocking solution (TBST-FBS 10%). Slides were washed in distilled water and incubated with primary antibodies against Ki67 (Ventana 790–4286) or hemicentin-1 (18837-1-AP, ProteinTech) overnight in a humidified chamber at 4˚C. Slides were washed 4x 5 minutes with TBST under gentle agitation and incubated with secondary antibodies (anti-mouse IgG AlexaFluor 488 (Invitrogen) or anti-rabbit IgG AlexaFluor 647 (Invitrogen), respectively). After 3 TBST washes for 5 minutes under gentle agitation, the nuclei were labelled with DAPI (2μM, 422801, Biolegend) for 15 minutes in the dark. Immunostaining was revealed using the HIGHDEF red IHC chromogen (AP) (Enzo Life Science), and imaged with a Microscope Slide Scanner (ZEISS Axioscan). For confocal microscopy, slides were washed 3x 5 minutes with TBST under gentle agitation, fixed with 8 μL of Prolong Glass Antifade Montant (ThermoFisher), and recorded with a Leica Stellaris device. All images were analyzed using the QuPath-0.3.0 software [76].

## Confocal microscopy and flow cytometry analysis of PBMCs

After venipuncture, PBMCs were purified by Percoll gradient centrifugation and cultured in RPMI 1640 (Lonza) supplemented with 10% decomplemented FBS, 1% of penicillin-streptomycin solution, phorbol 12-myristate 13-acetate (PMA, 10ng/mL, Invivogen) and ionomycin (1 μg/mL, VWR) [25,77]. After fixation in 4% PFA and permeabilization with PBS-Triton X-100 0.5%, cells were labelled with antibodies directed against p24 (4'G9 monoclonal) and hemicentin-1. After staining with Draq5 (Invitrogen), samples were incubated with appropriate conjugates (anti-mouse IgG AlexaFluor 488 and anti-rabbit IgG

AlexaFluor 647, respectively), mounted on slides using Fluoroshield (Sigma-Aldrich) and imaged with the Zeiss LSM 510 confocal microscope. Data was analyzed with the ZEN Blue software (Zeiss). For flow cytometry analyzes, cell events were recorded with a BD FACSAria (BD Biosciences) or a CytoFLEX (Beckman Coulter) cytometer. Data was analyzed using the BD FACS Diva (BD Biosciences), CytoFLEX (Beckman Coulter), FlowJow V10 and the CytExpert 2.5 softwares.

### Transcriptomic analysis

After Percoll gradient centrifugation, $10^6$ PBMCs were labelled with anti-sIgM antibody (monoclonal 1H4) and loaded onto a LS MACS column (Miltenyi Biotec) fixed on a Quadro-MACS magnetic separator (Miltenyi Biotec). After collecting the flowthrough fraction (*i.e.*, non-B cells) and washing the column with DPBS, the B cells were recovered. Total RNA was extracted from the B and non-B populations using the RNA Nucleospin extraction kit (Macherey-Nagel) according to the manufacturer's protocol. After verifying with a bioanalyzer that the RNA Integrity Number (RIN) exceeded 7.0, poly-A libraries were generated with the Illumina TruSeq stranded mRNA kit (Illumina) and sequenced with a NovaSeq6000 sequencing device (Macrogen). At least 30 million 100 bp paired-end, reads were recorded corresponding to a sequencing coverage of approximately 46 using the Lander/Waterman equation. After quality check with FastQC, reads were aligned on the ovine reference genome (Oar_rambouillet_v1.0) using HISAT2 (version 2.2.1). The number of sequences that mapped to known genes were quantified with the featureCounts method from subread package (version 2.0.3). A Principal Components Analysis (PCA) was performed to exclude sample outliers. Differentially expressed genes were identified using DESEq2 (version 1.32.0). Volcano plots of top significant genes were generated using the EnhancedVolcano (version: 1.16.0) and ggplot2 (version 3.3.3) R packages. Chord diagrams were generated using GOplot (version:1.0.2) R package.

### Statistics

All statistical analyses were performed using the GraphPad Prism 8.0.1 and the Microsoft Office 365 Suite softwares. For all tests, the threshold significance α was set at 0.05. Normality of distributions was tested according to the D'Agostino & Pearson test, using the method of Royston. Luciferase luminescence, number of syncytia, proviral loads, mean fluorescence intensity and number of positive cells in immunohistochemistry were compared between groups according to Mann-Whitney tests or unpaired t tests. Fold changes of RNA quantities were calculated according to two-tailed Wilcoxon signed rank test. Proviral load kinetics between WT and A1-infected animals were evaluated with Wilcoxon matched-pairs signed rank test. Kaplan-Meier survival curves were compared using the Log-rank Mantel-Cox test.

### Supporting information

**S1 Data. The A1 mutation decreases antisense transcriptional activity of the LTR in OVK cells.** Luciferase activities promoted by the minimal (R-U5, **A-B**) and full-length LTR (**C-D**) in OVK cells. Reporter plasmids (WT, A, A1, A2, B1) and vectors expressing either mock (pSG5) or the viral transactivator Tax (pSGTax) are described in **Fig 1B**. Twenty-four hours after transfection of OVK cells, Firefly (**A**) and Renilla (**B**) luciferase activities were measured and normalized to the mean luminescence of the pSGTax + pLTR samples, arbitrarily set to 100. RLU: relative luminescence units. Data results from at least three independent experiments. p-values were calculated according to Mann-Whitney tests.
(TIF)

**S2 Data. Antisense transcription is reduced in HEK293FT transfected with the A1 provirus and co-cultured with CC81 cells.** HEK293FT cells transfected with either pBLV344 (WT) or pBLV344-A1 (A1) were cocultured with CC81 cells for 24 hours. After extraction and RT-qPCR, *tax*, *as1-s/L* and *as2* RNAs were quantified by the ΔΔCT method using ß-actin as house-keeping gene. Data resulting from three independent experiments were normalized to the means of the WT samples, arbitrarily set to 1. p-values were calculated according to two-tailed Wilcoxon signed rank test.
(TIF)

**S3 Data. Volcano plots of most significant up/down-regulated genes in B-cells and non-B cells of sheep infected with mock, the A1 mutant or the wild-type virus.** Volcano plots were generated using the EnhancedVolcano (version: 1.16.0) and ggplot2 (version 3.3.3) R packages. The x and y axes correspond to $\log_2$(fold-change) (FC) and $-\log_{10}$(p-value), respectively. Cut-off values were defined as p-adj < 0.05 and |Log2FC| > 1. NI means not-infected and WT is wild-type.
(TIF)

**S4 Data. Differential Gene Expression analysis of B lymphocytes from sheep infected with the A1 mutant vs WT. (A)** Top 25 list of significantly upregulated genes ranked by $\log_2$(FC). **(B)** Top 25 list of significantly downregulated genes ranked by $\log_2$(FC).
(TIF)

**S5 Data. Differential Gene Expression analysis of B lymphocytes from sheep infected with the A1 mutant vs mock. (A)** Top 25 list of significantly upregulated genes ranked by $\log_2$(FC). **(B)** Top 25 list of significantly downregulated genes ranked by $\log_2$(FC).
(TIF)

**S6 Data. Differential Gene Expression analysis of B lymphocytes from sheep infected with the WT mutant vs mock. (A)** Top 25 list of significantly upregulated genes ranked by $\log_2$(FC). **(B)** Top 25 list of significantly downregulated genes ranked by $\log_2$(FC).
(TIF)

**S7 Data. Differential Gene Expression analysis of non-B cells from sheep infected with the A1 mutant vs WT. (A)** Top 25 list of significantly upregulated genes ranked by $\log_2$(FC). **(B)** Top 25 list of significantly downregulated genes ranked by $\log_2$(FC).
(TIF)

**S8 Data. Differential Gene Expression analysis of non-B cells from sheep infected with the A1 mutant vs mock. (A)** Top 25 list of significantly upregulated genes ranked by $\log_2$(FC). **(B)** Top 25 list of significantly downregulated genes ranked by $\log_2$(FC).
(TIF)

**S9 Data. Differential Gene Expression analysis of non-B lymphocytes from sheep infected with the WT mutant vs mock. (A)** Top 25 list of significantly upregulated genes ranked by $\log_2$(FC). **(B)** Top 25 list of significantly downregulated genes ranked by $\log_2$(FC).
(TIF)

**S10 Data. Splicing profiles of the viral RNAs in sheep PBMCs.** RNA-Seq data was aligned on the genomic RNA sequence of the BLV344 provirus using STAR. The jsdbOverhang19 parameter was set at 19. Sashimi plots were generated for each sheep infected with the pBLV344 (WT) or pBLV344-A1 (A1) provirus using the Integrative Genomic Viewer desktop application (IGV_2.18.2). The minimum junction coverage was set at 2. Splicing of the as1-S/L

RNA was detected for the three sheep inoculated with the WT. Due to low number of reads for the sheep inoculated with pBLV344-A1, the splicing pattern of the A1 virus was indeterminate.
(TIF)

**S11 Data. Hemicentin-1 protein accumulates in extracellular spaces.** Spleen **(A)** and lymph node **(B)** sections from WT and A1-infected sheep were stained with DAPI (nuclei in blue) and labelled with anti-HMCN1 antibody combined with an AlexaFluor 647 conjugate (in red). Slides were imaged with a Leica Stellaris confocal microscope.
(TIF)

**S12 Data. Pathogenic features of sheep inoculated with WT/A1 proviruses.** Sheep of the Vendéen breed were inoculated with mock or WT/A1 proviruses and examined at regular intervals to evaluate pathogenesis. Upon blood collection, PBMCs were separated by Percoll-based gradient centrifugation. The percentages of B-cell in the PBMCs were determined by flow cytometry using an anti-sIgM antibody (1H4). After DNA extraction, proviral loads (in copies /100 PBMCs) were quantified by qPCR. The death resulted from leukemia/lymphoma, causes unrelated to BLV infection (unknown) or euthanasia for biopsy samplings (end of experiment). The pictures illustrate normal and leukemic blood after PBMC centrifugation.
(TIF)

**S13 Data. RNA structure prediction of the 3' and 5' extremities of the WT/A1 genomic RNAs. (A)** Structure prediction for the R-U5 region of the WT RNA were made using the RNAfold WebServer by the Vienna RNA Websuite. The Primer Binding Site (PBS) is represented in purple. **(B)** Corresponding structure of the A1 RNA. The red arrow points to the A1 mutation. **(C)** Structure prediction of the U3-R region of the WT RNA. The Rex responsive element (RxRE) is represented in blue and the Polypurine tract (PPT) is colored in green. **(D)** Corresponding prediction of the A1 RNA folded structure.
(TIF)

**S14 Data. Co-transfection of pBLV344 (WT) or pBLV344-A1 does not affect luciferase activity of a pHMCN1-Luc reporter plasmid. (A)** The pHMCN1-Luc reporter plasmid contains a 1570bp fragment overlapping the *Hmcn1* gene promotor region (corresponding to the 64722376–64723946 location chromosome 12 of the Ovis Aries reference genome Oar_v4.0) cloned upstream of the humanized Luc2 Firefly luciferase gene. **(B)** Luciferase activities measured twenty-four hours after co-transfection of either pBLV344 or pBLV344-A1 plasmids. Firefly luciferase activities were normalized to the mean luminescence of the pSGTax samples, arbitrarily set to 1. Data results from of at least 10 independent experiments. p-values were calculated according to the Mann-Whitney test. RLU: Relative Luminescence Units. Ori: Origin of replication. AmpR: ampicillin resistance gene.
(TIF)

**S15 Data. List of primers used for site-directed mutagenesis, proviral load quantification and RT-qPCR.**
(TIF)

## Acknowledgments

We appreciate the valuable help of the GIGA technology platforms, in particular the imaging, viral vectors, genotranscriptomic and immunohistology facilities. We would like to thank Sui Xiukun and Lin Li for their contribution on site-directed mutagenesis. The pAsLuc(Fire)-HTLV-Luc(Reni) plasmid was kindly provided by Benoît Barbeau.

## Author Contributions

**Conceptualization:** Thomas Joris, Luc Willems.

**Data curation:** Thomas Joris, Lorian Gouverneur, Xavier Saintmard, Lea Vilanova Mañá, Majeed Jamakhani, Michal Reichert.

**Formal analysis:** Thomas Joris, Lea Vilanova Mañá.

**Funding acquisition:** Luc Willems.

**Investigation:** Thomas Joris, Thomas Jouant, Luc Willems.

**Methodology:** Thomas Joris, Lea Vilanova Mañá, Luc Willems.

**Project administration:** Luc Willems.

**Resources:** Jean-Rock Jacques, Majeed Jamakhani, Michal Reichert, Luc Willems.

**Software:** Thomas Joris, Lea Vilanova Mañá, Majeed Jamakhani.

**Supervision:** Luc Willems.

**Validation:** Thomas Jouant, Lea Vilanova Mañá, Luc Willems.

**Visualization:** Thomas Joris, Lea Vilanova Mañá, Majeed Jamakhani.

**Writing – original draft:** Thomas Joris, Luc Willems.

**Writing – review & editing:** Thomas Joris, Thomas Jouant, Luc Willems.

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
