## [Decision Letter · Decision Letter 0]

22 Mar 2024

Dear Dr Joris,

Thank you very much for submitting your manuscript "Antisense transcription of bovine leukemia virus is essential for viral replication and oncogenesis" for consideration at PLOS Pathogens. As with all papers reviewed by the journal, your manuscript was reviewed by members of the editorial board and by several independent reviewers. In light of the reviews (below this email), we would like to invite the resubmission of a significantly-revised version that takes into account the reviewers' comments.

All three reviewers found your results of interest and importance. However, they also raise several points that require either further experiments or significant clarification. Reviewer 1 identifies a potentially important additional experiment; the experiments suggested by reviewer 3 are 'minor' and are optional.

We cannot make any decision about publication until we have seen the revised manuscript and your response to the reviewers' comments. Your revised manuscript is also likely to be sent to reviewers for further evaluation.

Sincerely,

Charles R M Bangham, ScD FRS

Academic Editor

PLOS Pathogens

Richard Koup

Section Editor

PLOS Pathogens

Michael Malim

Editor-in-Chief

PLOS Pathogens

orcid.org/0000-0002-7699-2064

All three reviewers found your results of interest and importance. However, they also raise several points that require either further experiments or significant clarification. Reviewer 1 identifies a potentially important additional experiment; the experiments suggested by reviewer 3 are 'minor' and are optional.

Reviewer's Responses to Questions

**Part I - Summary**

Reviewer #1: In the present study, Joris et al. reported that a mutation in the BRE impaired antisense transcriptions of BLV provirus, resulting in favorable survival of BLV-infected sheep without lymphoma/leukemia. This study is well-concepted and done in vitro and in vivo to evaluate their findings. However, there are still major concerns to be discussed more as mentioned below.

Reviewer #2: The authors investigated whether the antisense transcript from the BLV genome plays a role in viral persistence and pathogenesis. They utilized a molecular BLV clone with reduced antisense transcription due to nucleotide mutations, referred to as the A1 mutation, in the 3’LTR promoter region. The authors confirmed that these mutations reduced antisense transcription levels, but not sense transcription, through a reporter assay in vitro.

Subsequently, the authors demonstrated that recombinant BLV with the A1 mutation exhibited a lower expression level of antisense transcription but showed similar cell fusion activity to the wild-type virus in in vitro assays. The analysis was further extended to an in vivo infection model. In infected sheep, the antibody levels against the viral envelope protein appeared similar, but the PVL significantly differed between the wild-type and A1 mutant BLV. Antisense RNA and p24 capsid protein levels were decreased in A1 BLV-infected sheep.

To analyze the underlying mechanism of the observed phenotype in infected sheep, the authors conducted RNA-seq analysis using B cells from wild-type or A1 mutant BLV-infected sheep. The authors focused on the Hmcn-1 gene, which was highly upregulated in B cells from A1 BLV-infected sheep. Finally, the authors demonstrated lower viral persistence and pathogenesis in the A1 BLV mutant virus compared to the wild-type BLV.

The study provides evidence regarding the roles of antisense transcription from the BLV genome. It offers valuable insights into how BLV achieves persistent infection in the host, sometimes resulting in pathogenesis. However, before considering the paper for publication, several points need revision. Firstly, the flow of the writing feels disjointed; consider revising transitions between paragraphs for a smoother and more logical progression of ideas. Secondly, grammatical errors and awkward phrasing are present throughout the paper. It is crucial to carefully proofread and edit the manuscript for clarity and precision. Thirdly, the focus of the paper appears unclear. Please refine and clarify the research question and the mechanism underlying the observed phenotype in this study.

Reviewer #3: This paper directly demonstrates the impact of BLV antisense transcription on leukemogenesis by combining elaborate in vitro and dynamic in vivo experiments using infectious molecular clones of mutant strains. The strategy and methodology of the study are well designed, and the interpretation of the results is clear without being excessive. I request that two minor additional experimental data be presented, but they would not affect the main conclusions of this paper. Although the detailed molecular mechanisms supporting the phenomena observed here remain unclear, the comparative virological approach using a sheep model is highly original and will be of interest to readers. The discussion is concisely written with selected key issues, but I would like to see minor corrections and additions, as some points seem insufficiently explained to me.

**Part II – Major Issues: Key Experiments Required for Acceptance**

Reviewer #1: 1. Regarding luciferase assays with a series of mutations for both UTR (Fig.1), promoter activity of the full-length LTR was assayed with A1 mutation only (Fig. 1F&G). To confirm the findings with R-U5, the other mutants should be evaluated similarly in the full-length LTR assay. In addition, in OVK cells, when pSGTax was used (Supple Fig1.C&D), no significant difference was found in both LTR assays. The authors need to discuss why the difference was observed between 293FT and OVK cells.

Also, the use of the name “A1” mutation should be avoided in the abstract. The naming of this mutant is used by the authors only in the paper and should not be a commonly known mutant name.

2. In terms of Fig.2B-2C, did co-cultured cells show reduced antisense transcriptions similarly to 293FT cells?

3. In Fig.5, although PVLs were decreased in both lymph nodes and spleen in the infected sheep with A1 mutant (Fig. 5C&D), Ki67 results were different in lymph nodes and spleen (Ki67 was more elevated in spleen with A1 mutant). Only this finding is at odds with the other results.

The finding of decreased PVL but increased proliferative capacity in the spleen needs more detailed study. The authors also discuss the correlation between BLV and HTLV-1. However, as is well-known, the aggressive phenotype of ATL markedly increases the amount of HTLV-1 provirus compared to carriers and chronic/smoldering ATL.

4. In terms of survival analysis (Fig. 5H), are all observed events in wild-type infected sheep the development of lymphoma or leukemia?

Reviewer #2: 1. Impact of A1 mutation of the BLV transcriptome.

It is not so clear why A1 mutation affects antisense but not sense transcription. Luciferase assay support the idea, but it is not clear if that is the case as whole provirus.

a. More information about how they selected and introduce certain pattern of mutations (Figure 1B)

b. Viral transcriptome pattern from with RNA-seq data can be shown on IGV. That would be useful information if A1 mutation would change splicing pattern of proviral transcript.

2. Some fundamental data is missing in sheep experiment.

I do agree and respect the value of data obtained from in vivo sheep model. However, there are several fundamental data is missing, so it was difficult to interpret the data.

a. How did they normalize dose of virus to inoculate each sheep?

b. How was the lymphocytosis in wild type and A1 mutant infected sheep?

c. They enriched B cells for RNA-seq analysis. How was the proportion of B cells in the PBMCs?

d. Figure 3B; They showed PVL as copy number/ug DNA. I recommend to show copy number/100 PBMCs as usual.

e. Figure 3C; How did they normalize RNA loads? PBMCs or B cells or infected cells?

Since PVL was very low in A1 sheep, as2 transcripts of A1 sheep may be higher than that of wild type if we normalized by PVL.

f. Figure 3E; It is good to know percentage of p24 positive cells. Then we can know % of infected cells in the PBMCs. Then we can have some idea how many copy of provirus per infected cells in this sheep model.

g. Figure 3G; It is informative if they show sense and antisense BLV transcript level in this plot.

h. Figure 4A & B; it would be much informative if the authors show IHC staining of B cell marker.

i. Figure 4E: What total means in the calculation?

3. Significance and mechanism of Hmcn-1 upregulation in B cells in A1-BLV-infected.

a. Figure 3H; To interpret this data, again it is useful to know what was the percentage of A1 BLV infected cells in the B cells. If infectivity is very low, the effect of A1 mutation on B cell transcriptome should not be mediated by cell intrinsic way but indirect effect. Current authors’ explanation is cell intrinsic mechanisms. This should be supported by high infectivity in B cells. It would be more convincing if the authors would show effect of antisense BLV transcript on Hmcn-1 expression.

b. It is not clear how Hmcn-1 expression change induced phenotypic changes between wild type and A1 mutant BLV. If I understand correctly, logical flow is like below.

A1 mutation -> decreased BLV antisense RNA -> increased Hmcn-1 expression in B cell -> low viral persistence and pathogenesis

I’d like to ask authors to make the logic clear.

Reviewer #3: Two minor additional experiments are proposed, but I suggest that the experimental data presented in the current draft are sufficient for the main conclusions of this paper.

Figure 5B & 5A (L237): Additional data that Ki-67+ cells are T cells rather than B cells would be desirable. In general, a higher provirus level in the early stages of infection seems to lead to a stronger T-cell response. However, the result that the cell proliferation in spleen is rather induced in the A1 group with low provirus levels seems to contradict this.

Figure 5G (L249): If DNA samples are available, PVL data up to the end of the study period should be shown. Some individuals not marked dead (cross) (especially group A1) have PVL data ending in the middle of the study period.

**Part III – Minor Issues: Editorial and Data Presentation Modifications**

Reviewer #1: Page 7, Line 136

“OVK” should be spelled out and explained when it is used firstly in the text.

Page 10, Line 198

“analyzes” should be an error in writing.

Page 11, Line 217

For Fig. 4F, incucyte live-cell imaging should be described in more detail in the text to make it easier for readers to understand.

Reviewer #2: 1. Figure 1F, Figure 2C; It is not well explained why A1 mutation showed different effect on sense transcription. If they show more control data about transfection efficiency is similar between wild type and A1 mutant, the data will be more convincing.

2. Figure 2C and 2D; How can we explain why Tax expression was low but infectivity was high in A1 mutant?

3. Line 55; Please correct sentence “Gene expression was affected..”

4. Line59-60; The author mentioned ‘These findings may offer valuable insights into understanding the relevance of antisense transcription in the context of human T-cell leukemia virus (HTLV-1).’ There are similarities as well as difference between HTLV-1 and BLV. I am wondering what is the exact value of this research for HTLV-1 infection?

5. There are many typo and grammar issues

6. Figure 4; image is not so clear. Please use better ones.

7. Figure 4E; The phenomenon we are discussing in this study is In B cells. Why the authors use OVK cells?

8. Line 240; Again, it is not clear how the authors performed special distribution of provirus. What is the total PVL value. Did they extract genomic DNA from whole body and calculated total PVL in the sheep?

9. Figure 5; how many sheep they analyzed?

10. Line 288; Is this sentence correct?

11. Line 656; No description of Incucyte live imaging in the text/methods

Reviewer #3: L207: Regarding the cellular localization of Hemicentin-1 (Figure 4B), it seems that A1-infected PBMCs have a stronger localization to the plasma membrane than A1-infected PBMCs. If this is correct, what is the link between the change in localization and the effect on cell migration?

L249: In Figure 5G, there are some individuals whose plots are interrupted in the middle of the study, but according to Figure 5H, they survived until the end of the study. Is it correct? Since there are discrepancies in the data interpretation, please provide a list of the animal information (animal ID, breed, sex, age, date and cause of death or withdrawal from the study, clinical and necropsy findings) as Supplementary Data., then identify the individuals on the graphs (Figure 5F and G).

L250: Is it confirmed that the cause of death of all sheep that died during the observation period was due to the onset of a tumor? If not, please indicate the incidence of tumors.

L258: I strongly agree that a comparative virological approach using a BLV-infected sheep model would be beneficial in understanding the pathogenesis of HTLV-1-related disease. However, whether the findings on BLV antisense transcription in this study can be extrapolated to disease mechanisms induced by antisense transcription of HTLV-1 needs careful validation and discussion. In a previous study by the authors (Ref. 23), they reported that HTLV-1 HBZ RNA inhibits 5'LTR promoter activity and abolishes proviral sense transcription, thus facilitating HTLV-1 entry into latent infection and immune evasion. This is at odds with the results obtained in this study for BLV and does not allow us to consider the same RNA-dependent epigenetic mechanism of antisense transcription of BLV and HTLV-1. How about adding a little more on this point in the discussion?

L303: As discussed by the authors, they do not have a detailed explanation of why antisense transcription suppresses BLV replication in vivo. The authors propose three hypotheses about it (L303-306). A more detailed explanation of each hypothesis would be helpful for the reader's understanding.

L316: The author indicate that TGFβ and MAPK may be involved in the induction of Hemicentin-1 expression in B cells. Please indicate references if any.

L660 or Figure 4H: Please explain in the legend which cells you have determined as Hemicentin-1 positive cells or mark the immunostaining pictures.

PLOS authors have the option to publish the peer review history of their article (what does this mean?). If published, this will include your full peer review and any attached files.

Reviewer #1: No

Reviewer #2: **Yes: **Yorifumi Satou

Reviewer #3: No
---

## [Decision Letter · Decision Letter 1]

12 Oct 2024

Dear Mr Jouant,

We are pleased to inform you that your manuscript 'Reduction of antisense transcription affects bovine leukemia virus replication and oncogenesis' has been provisionally accepted for publication in PLOS Pathogens.

Best regards,

Charles R M Bangham, ScD FRS

Academic Editor

PLOS Pathogens

Richard Koup

Section Editor

PLOS Pathogens

Michael Malim

Editor-in-Chief

PLOS Pathogens

orcid.org/0000-0002-7699-2064

Thank you for submitting the revised version of your paper. All reviewers are now very positive in their approval of the revised manuscript.

Reviewer Comments (if any, and for reference):

Reviewer's Responses to Questions

**Part I - Summary**

Reviewer #1: All my concerns have been solved. I hope you would uncover the discrepancy where we discussed about Fig 5C&D in the near future.

Reviewer #2: Thank you for submitting the revised version of your manuscript. I appreciate the careful and thorough responses to the initial feedback. The revised manuscript is much improved, with significant enhancements in both clarity and structure.

The overall flow of the paper is now smooth, and the transitions between paragraphs and sections provide a more coherent narrative. The grammatical and phrasing issues have been addressed appropriately, making the text much clearer and more precise. Furthermore, the research question and mechanism underlying the observed phenotype are now well articulated, and the focus of the study is much clearer.

The study provides valuable insights into the role of antisense transcription in BLV persistence and pathogenesis, supported by rigorous in vitro and in vivo experimental data. The RNA-seq analysis and its connection to the upregulation of Hmcn-1 in A1 BLV-infected sheep is an interesting finding that enhances the depth of the study.

Reviewer #3: This paper directly proves the impact of BLV antisense transcription on the leukemogenesis based on a well-designed research strategy and methodology. I think the comparative virological approach using a sheep model is highly original and will be of interest to readers. In the revised manuscript, the results and discussion have been further enhanced, and I expect that this revision will promote readers' understanding without excess or deficiency.

**Part II – Major Issues: Key Experiments Required for Acceptance**

Reviewer #1: (No Response)

Reviewer #2: No issue

Reviewer #3: The revisions made in this round adequately address the requests made by the reviewers, and that no further major revisions are necessary.

**Part III – Minor Issues: Editorial and Data Presentation Modifications**

Reviewer #1: (No Response)

Reviewer #2: No issue

Reviewer #3: No minor modifications are necessary.

PLOS authors have the option to publish the peer review history of their article (what does this mean?). If published, this will include your full peer review and any attached files.

Reviewer #1: **Yes: **Kosuke Toyoda

Reviewer #2: **Yes: **Yorifumi Satou

Reviewer #3: **Yes: **Tomohiro Okagawa

---

## [Editor Report · Acceptance letter]

25 Oct 2024

Dear Mr Jouant,

We are delighted to inform you that your manuscript, "Reduction of antisense transcription affects bovine leukemia virus replication and oncogenesis," has been formally accepted for publication in PLOS Pathogens.

Best regards,

Michael Malim

Editor-in-Chief

PLOS Pathogens

orcid.org/0000-0002-7699-2064